# Pistachio Consumption Alleviates Inflammation and Improves Gut Microbiota Composition in Mice Fed a High-Fat Diet

**DOI:** 10.3390/ijms21010365

**Published:** 2020-01-06

**Authors:** Simona Terzo, Flavia Mulè, Gaetano Felice Caldara, Sara Baldassano, Roberto Puleio, Maria Vitale, Giovanni Cassata, Vincenzo Ferrantelli, Antonella Amato

**Affiliations:** 1Department of Experimental Biomedicine and Clinical Neuroscience (BioNec), University of Palermo, Via del Vespro 129, 90127 Palermo, Italy; simona.terzo01@unipa.it; 2Department of Biological- Chemical- Pharmaceutical Science and Technology (STEBICEF), University of Palermo- Viale delle Scienze, Edificio 16, 90128 Palermo, Italy; flavia.mule@unipa.it (F.M.); opba@unipa.it (G.F.C.); sara.baldassano@unipa.it (S.B.); 3Istituto Zooprofilattico Sperimentale della Sicilia “A. Mirri”, Via Gino Marinuzzi 3, 90129 Palermo, Italy; roberto.puleioizs@gmail.com (R.P.); maria.vitale@izssicilia.it (M.V.); giovanni.cassata@izssicilia.it (G.C.); vincenzo.ferrantelli@izssicilia.it (V.F.)

**Keywords:** obesity-related inflammation, pistachio intake, gut microbiota, HFD mice, adipose tissue

## Abstract

High-fat diet (HFD) induces inflammation and microbial dysbiosis, which are components of the metabolic syndrome. Nutritional strategies can be a valid tool to prevent metabolic and inflammatory diseases. The aim of the present study was to evaluate if the chronic intake of pistachio prevents obesity-associated inflammation and dysbiosis in HFD-fed mice. Three groups of male mice (four weeks old; *n* = 8 per group) were fed for 16 weeks with a standard diet (STD), HFD, or HFD supplemented with pistachios (HFD-P; 180 g/kg of HFD). Serum, hepatic and adipose tissue inflammation markers were analyzed in HFD-P animals and compared to HFD and STD groups. Measures of inflammation, obesity, and intestinal integrity were assessed. Fecal samples were collected for gut microbiota analysis. Serum TNF-α and IL-1β levels were significantly reduced in HFD-P compared to HFD. Number and area of adipocytes, crown-like structure density, IL-1β, TNF-α, F4-80, and CCL-2 mRNA expression levels were significantly reduced in HFD-P subcutaneous and visceral adipose tissues, compared to HFD. A significant reduction in the number of inflammatory foci and IL-1β and CCL-2 gene expression was observed in the liver of HFD-P mice compared with HFD. *Firmicutes*/*Bacteroidetes* ratio was reduced in HFD-P mice in comparison to the HFD group. A pistachio diet significantly increased abundance of healthy bacteria genera such as *Parabacteroides*, *Dorea*, *Allobaculum*, *Turicibacter*, *Lactobacillus*, and *Anaeroplasma*, and greatly reduced bacteria associated with inflammation, such as *Oscillospira*, *Desulfovibrio*, *Coprobacillus*, and *Bilophila*. The intestinal conductance was lower in HFD-P mice than in the HFD mice, suggesting an improvement in the gut barrier function. The results of the present study showed that regular pistachio consumption improved inflammation in obese mice. The positive effects could be related to positive modulation of the microbiota composition.

## 1. Introduction

Obesity and overweight in western societies and developing countries has become one of the most important public health problems. These, in part, result from the consumption of unbalanced hypercaloric diets that cause excessive visceral fat accumulation [1]. Obesity is associated with chronic low-grade inflammation, which can impair glucose and fatty acid metabolism, leading to insulin resistance and metabolic syndrome [2]. Most studies have focused on adipocytes as the source of inflammatory mediators in this pathology. Storage of excess of triacylglycerol induces hyperplasia and hypertrophy of the adipocytes with altered release of adipokines and pro-inflammatory cytokines, which in turn enhance the recruitment of immune cells, especially macrophages [3]. Therefore, the macrophages in obese adipose tissues are considered to be a major source of pro-inflammatory cytokines, such as TNF-α and IL-6, which are involved in abnormal metabolisms [4]. 

However, recent studies have suggested that changes in the composition of the gut microbiota might be associated with the development of metabolic disorders related to obesity [5,6,7]. Indeed, a diet that is rich in saturated fat and poor in fiber is responsible for weight gain, changes in gut microbiota [8], and increased intestinal permeability [9]. The intestinal barrier dysfunction causes an increased circulation of lipopolysaccharides (LPS) derived from gram-negative bacteria [10,11]. In turns, LPS spread participates in metabolic endotoxemia development, adipose tissue dysfunction, and systemic inflammation, triggering obesity-related complications [12]. 

Nutritional strategies can represent a valid support to prevent metabolic and inflammatory diseases. Increased consumption of fruit and vegetables could prevent chronic diseases such as cardiovascular disease and could prevent body weight gain [13]. Additionally, plant-based foods reduce metabolic syndrome risk [14]. Functional food, i.e., food that can modulate the richness and biodiversity of the gut microbiota and consequently induce a healthier metabolic status, has received increased attention from researchers worldwide [15,16]. It is widely accepted that the consumption of nuts such as almonds, walnuts, and pistachios, as a part of the daily diet provides beneficial effects on human health [17]. Among nuts, Pistachio (*Pistacia vera* L.) is the healthiest due to its fatty acid composition and bioactive compound content (such as lutein and anthocyanin) [18,19]. In recent years, the anti-inflammatory effects of pistachios and the anti-inflammatory activity of its components have been the object of numerous studies. In particular, the anti-inflammatory effects have been reported in both in vitro models [20,21] and in various animal models [22,23,24]. The antimicrobial properties of polyphenolic fractions obtained from roasted pistachios have also been demonstrated [25,26]. 

Moreover, we have already shown that the daily pistachio intake prevents and improves some obesity-related metabolic dysfunctions such as dyslipidemia and hepatic steatosis in mice with diet-induced obesity, through a positive modulation of lipid-metabolizing gene expression [27]. Nevertheless, no study has characterized the links between pistachio supplementation, adiposity-related inflammation, and gut microbiota alterations. High-fat diet (HFD) mice are considered a good obese model to characterize the beneficial potential of various treatments on obesity-related disorders since they develop dyslipidemia, hyperglycemia [28,29], type 2 diabetes mellitus [30], hepatic steatosis [31], atherosclerosis [32], and neurodegeneration [33]. 

Therefore, the purpose of the present study was to investigate whether chronic pistachio consumption is able to prevent the associated visceral–obesity inflammation, the altered composition of gut microbiota, and the intestinal barrier integrity in HFD-obese mice.

## 2. Results

### 2.1. Impact of Pistachio Consumption on Body Weight and Metabolic Parameters

As previously reported [27,31], after 16 weeks on HFD, mice showed a significant increase in body weight, triglyceride, and cholesterol plasma concentration in comparison with the standard diet (STD)-fed lean animals. In HFD supplemented with pistachio (HFD-P)-fed mice, triglyceride and cholesterol concentrations were significantly reduced, in comparison with untreated obese mice, whereas the body weight and food intake were similar (Table 1). 

### 2.2. Impact of Pistachio Consumption on TNF-α and IL-1β Expression

To examine whether pistachio consumption prevents the systemic inflammation induced by HFD, the serum levels of the pro-inflammatory cytokines TNF-α and IL-1β were evaluated by ELISA. As shown in Figure 1, intake of pistachios significantly decreased the HFD-induced high levels of IL-1β and TNF-α.

### 2.3. Impact of Pistachio Consumption on Adipocytes Hypertrophy

Adipocyte area (μm^2^) and adipocyte size distribution (%) were analyzed in visceral adipose tissues (VAT) and subcutaneous adipose tissues (SAT). The adipocytes area in the HFD was significantly higher than that in the lean group; however, the degree of increase was significantly suppressed by HFD-P suggesting that pistachio chronic intake reduces the hypertrophy in both fat depots examined (Figure 2A–C).

### 2.4. Impact of Pistachio Consumption on Adipose and Hepatic Tissue Inflammation

The presence of Crown Like Structures CLS as an index of macrophage infiltration was evaluated and quantified in VAT and SAT. As shown in Figure 3, more crown-like structures were detected in HFD mice, as compared to the lean animals. Interestingly, in HFD-P mice, the CLS density was significantly lower in comparison to the HFD adipose tissues (Figure 3A,B). Furthermore, RT-PCR analysis revealed significantly higher levels of IL-1β, TNF-α, F4-80, and CCL2 mRNA in HFD mouse VAT and SAT than in the lean mice. However, pistachio-diet reduced the increase of the pro-inflammatory cytokines and the macrophage infiltration markers in both adipose tissue depots (Figure 3C). 

As previously reported [27], pistachio consumption counteracted the hepatic steatosis development consequent to HFD (Figure 4A). HFD mice showed higher infiltration of inflammatory cells in the liver compared to the STD animals. Nevertheless, infiltration was reduced in HFD-P livers in comparison to the HFD ones (Figure 4A,B). Moreover, pistachio intake significantly prevented the increase in hepatic mRNA levels of IL-1β and CCL2 observed in the HFD liver, as compared to the STD animals (Figure 4C).

### 2.5. Impact of Pistachio Consumption on Gut Microbial Community

To examine the changes of the gut microbiota in response to the pistachio diet in obese HFD mice, we analyzed the microbial composition in the feces of mice fed STD, HFD, and HFD-P, through Next-Generation Sequencing (NGS) analysis. After 16 weeks of HFD feeding, a decrease in the phyla *Bacteroidetes* and an increase in the phyla *Firmicutes* and *Proteobacteria* relative to STD were observed both in the HFD group and in the HFD-P mice (Figure 5A). The ratio of *Firmicutes* to *Bacteroidetes* was significantly higher in the HFD group than in the lean mice group, consistent with the microbial changes of the two phyla in the mice with HFD-induced obesity. Although this value was also an index of dysbiosis in the HFD-P group, it was significantly improved by pistachio intake (Figure 5B). Interestingly, *Tenericutes* abundance of the HFD-P mice was significantly increased in comparison with the HFD control mice; on the contrary, the pistachio diet significantly reduced the *Proteobacteria* abundance (Figure 5A). 

At the genus level, a pistachio diet significantly altered the abundances of 10 genera in a positive direction, as compared to the HFD animals. In particular, an abundance of *Parabacteroides*, *Dorea*, *Allobaculum*, *Turicibacter*, *Lactobacillus*, and *Anaeroplasma* genera was observed, while *Oscillospira*, *Desulfovibrio*, *Coprobacillus*, and *Bilophila* abundance was reduced in comparison to the HFD mice (Figure 6).

### 2.6. Impact of Pistachio Consumptiom on the Intestinal Barrier

Barrier integrity in small intestine sections was evaluated in an Ussing chamber system through conductance measurements of mucosal preparations from all experimental groups. Conductance values in the duodenal sections from animals fed HFD were significantly higher than those from the STD group (about 60% increase). Notably, the conductance values from the HFD-P group were significantly lower than the HFD group and were very similar to the lean group (Figure 7).

## 3. Discussion

The present study provided evidence that regular pistachio intake in HFD-fed obese mice ameliorates systemic and metabolic tissue inflammation, positively modulates the gut microbial composition, and increases the intestinal barrier function. 

Previous in vitro and in vivo studies have examined the antioxidant, anti-inflammatory, and anti-apoptotic potential of pistachio [21,34,35,36,37,38]. In particular, the pistachio properties were tested on carrageenan or LPS-induced acute inflammatory response [39,40], inflammatory bowel disease, colitis [24,41,42,43], cancer [44,45,46], and allergic inflammation in the asthmatic model [47]. To our knowledge, the present study was the first to explore the anti-inflammatory effects of pistachios in mice with HFD-induced obesity.

Obesity is characterized by a chronic low-degree inflammation. In fact, excessive calorie intake increases fat accumulation and the lipotoxicity activates the production of cytokines and the cells involved in innate immunity. This production promotes a chronic, low-grade inflammatory status, induces recruitment, and activation of mature immune cells and other cells, such as macrophages and adipocytes, respectively, which modify the tissue and reinforce the inflammatory response [12,48]. 

We previously reported that a pistachio-based diet exerts beneficial effects in HFD obese mice. In fact, it reduces the dyslipidemia and hepatic steatosis, and is able to prevent and improve visceral fat mass accumulation in HFD mice through a redistribution towards the subcutaneous fat depot, which is indicative of a healthier profile [27]. The present work not only confirms that the pistachio diet modifies fat depots, as suggested by the morphological analysis of visceral and subcutaneous adipose tissue, but also reduces the obesity-linked inflammatory status. 

First, we highlighted that a pistachio diet significantly prevents the increase of pro-inflammatory cytokines, TNF-α, and IL-1β induced by HFD in the systemic circulation. Furthermore, we provided evidence that visceral and subcutaneous adipose tissue and liver inflammation induced by obesity were strongly prevented by pistachio intake. Various inflammatory mediators are involved in adipose tissue and liver inflammation. In the adipose tissue, a paracrine loop linking fatty acids, TNF-α and CCL2 establishes a vicious cycle between adipocytes and macrophages that aggravates inflammation [46]. In the liver, the increased influx of fatty acids induces lipotoxic injury and activation of inflammatory response. Accordingly, an abundant expression of pro-inflammatory cytokines IL-1β and TNF-α is often associated with Non-Alcoholic Fatty Liver Disease (NAFLD) [49].

We found that HFD-P mice exhibit lower levels of TNF-α, F4-80, and CCL2 as well as minor macrophage infiltration, detected as CLS density in adipose tissues, in comparison to HFD animals. Additionally, in the liver, we found a reduction of IL-1β and CCL2 mRNA levels and a decreased number of inflammatory foci, in comparison to the HFD mice. These changes would favor an anti-inflammatory microenvironment that is able to counteract the biochemical dysfunctions occurring in adipose tissue or in the liver of HFD mice.

Obesity and metabolic disorders are complex processes that also involve crosstalk between the gut microbiota and host metabolism [50]. The gut microbiota might induce inflammation in visceral adipose tissue via the LPS and TLR4 signaling pathways with an increased macrophage infiltration and release of a variety of pro-inflammatory mediators, which in turn recruit additional macrophages to further propagate the chronic inflammatory status [51,52]. Therefore, in attempt to elucidate an eventual contribution of the gut microbiota to the beneficial pistachio effects, we investigated the profiling changes of the gut microbiota composition in mice, performing 16S rDNA sequencing through NGS analyses. Indeed, pistachio consumption can modify human gut microbiota composition by increasing the number of potentially beneficial butyrate-producing bacteria [53]. Our results demonstrated that the microbial communities were influenced by different type of diets. Analysis at the phylum level indicated that the fecal microbiota was dominated by seven major phyla: *Firmicutes*, *Bacteroidetes*, *Proteobacteria*, *TM7*, *Deferribacteres*, *Actinobacteria*, and *Tenericutes*. We observed a dramatic reduction in *Bacteroidetes* abundance and a marked increase in *Firmicutes*, in the HFD group, in accordance with the increased *Firmicutes* to *Bacteroidetes* ratio identified in obese humans and mice [54,55]. However, although the pistachio diet failed to maintain the *Firmicutes/Bacteroidetes* proportion observed in STD mice, the ratio value in HFD-P was significantly lower than the HFD group, suggesting a pistachio protective effect against dysbiosis. 

Interestingly, compared to the HFD control mice, we found a significant increase in the *Tenericutes* abundance and a significant decrease in *Proteobacteria* abundance in the HFD-P mice. Bacteria from the *Tenericutes* phylum have been found to be positively associated with the modulation of the immune system induced by high-polyphenol content food such as cocoa [56], and lower counts of these bacteria were found in the intestinal inflammation induced by dextran sodium sulfate [57]. Therefore, a more relative abundance of *Tenericutes* induced by the pistachio diet could provide some beneficial effects in the intestinal integrity. In addition, several reports have endorsed an abundance of *Proteobacteria* in the gut microbiota as a potential marker for obesity-related metabolic disorders in both humans and rodents [6,58]. Therefore, the lower level of *Proteobacteria* in the HFD-P-fed mice than the HFD mice could be indicative of less severe health conditions. 

At the genus level, *Lactobacillus* was significantly increased in the HFD-P, in comparison to the other groups. The relative abundance of *Lactobacillus* caused by pistachio intake can be interpreted as a positive effect because *Lactobacillus* is a well-known probiotic that has been associated with reduced colitis in several models of inflammatory bowel diseases [59], and has been shown to have protective effect in the intestinal barrier function and steatosis [60,61,62]. 

Interestingly, pistachio intake was found to improve the abundance of other genera that are usually associated with a positive impact on host health, such as *Parabacteroides*, *Dorea*, *Allobaculum*, *Turicibacter*, and *Anaeroplasma*. *Parabacteroides* is a genus predominantly found in the gut of healthy individuals, which is negatively correlated with body weight gain, liver steatosis, and epididymal fat accumulation [63]. The *Allobaculum* genus has been associated with a better mucus layer in the colon [64], suggesting that its decrease reflects the alteration of the mucus layer in HFD. Thus, a pistachio diet might prevent this alteration. Moreover, *Allobaculum* and *Dorea* are among the major producers of butyrate, an important fuel for epithelial colonocytes that have been shown to help maintain normal differentiation. Thus, an increase in the amount of butyrate generated in the gut might be an indication of improved health. Accordingly, butyrate-producing probiotics reduce NAFLD progression in rats [65] and attenuate HFD-induced steatohepatitis in mice, by improving intestinal permeability [66,67,68,69]. 

The decreased *Turicibacter* abundance in HFD mice, which was prevented by pistachio intake, fits well with previous data showing a depletion of *Turicibacter* in animal models of inflammatory bowel disease and confirms the hypothesis that *Turicibacter* is an anti-inflammatory taxon [70,71,72]. Recent data report that *Anaeroplasma* abundance is significantly decreased in obese rats, while the increased abundance is related to a reduction in fat accumulation and expression of inflammatory factors in the liver [73].

Another interesting effect of pistachio intake on gut microbiota concerns the decrease of genera associated with inflammation, such as *Desulfovibrio*, *Coprobacillus*, *Oscillosphira*, and *Bilophila*. *Desulfovibrio* is a genus responsible for 60% of the total hydrogen sulfide (H_2_S) production in the colon. H_2_S inhibits the mitochondrial respiration of colonic epithelial cells [74], reducing the diffusion of oxygen, and then subtracting energy that is useful to the beta-oxidation of butyrate [75]. Thus, it is likely that the reduction of H_2_S-producing bacteria by pistachio enhances the output of short chain fatty acids (SCFAs), such as butyrate, improving intestinal health and inflammation [76]. *Coprobacillus* has been reported to be negatively correlated with most of the features of obesity in obese rats [77,78]. An abundance of *Oscillospira* has been associated with systemic inflammation and altered intestinal permeability [79,80] and diets rich in polyphenols improve HFD-induced liver steatosis by reducing *Oscillospira* abundance [81,82]. *Bilophila* abundance seems to be related with colon inflammation [82]. A recent work reports that the treatment with phenolic compounds alleviate obesity-related inflammation in HFD-mice, by inhibiting the expansion of the *Bilophila* bacteria genus [76]. 

The changes in microbiota composition might be due to the different components of the pistachios, such as fatty acids, flavonoids, or fiber. Pistachios might exhibit prebiotic effects by enriching potentially beneficial microbes, such as lactic acid bacteria.

Therefore, taken together, these results suggest that the gut microbial alterations observed in HFD-P mice might be associated with pistachio metabolic and anti-inflammatory benefits. 

It is interesting to note that an increased intestinal conductance was observed in the small intestine of HFD mice, in comparison with lean or HFD-P mice, suggesting that HFD induces a decrease in the intestinal epithelial integrity and an increased ability of ions and small molecules to permeate through the paracellular pathway. According to our data, several studies report an increased gut permeability in the HFD mice [83,84]. The intestinal conductance value in the HFD-P group was similar to the lean group, suggesting that the pistachio diet is able to prevent an increase in permeability and, thus, showing the protective action of the pistachio diet on the intestinal barrier functions. Cani and collaborators [85] provided evidence that the development of metabolic endotoxemia and the linked metabolic disorders induced by high-fat feeding are associated with an increased intestinal permeability. Therefore, it is likely that the modulation of gut bacteria associated with increased intestinal barrier functions are involved in the anti-inflammatory effects of pistachio diet.

## 4. Materials and Methods

### 4.1. Animals and Diets

The procedures were performed in accordance with the conventional guidelines for animal experimentation (Italian D.L. No. 26/2014 and subsequent variations) and the recommendations of the European Economic Community (2010/63/UE). The experimental protocols were approved by the animal welfare committee of the Istituto Zooprofilattico Sperimentale della Sicilia “A. Mirri” (Palermo, Italy) and authorized by the Ministry of Health (Rome, Italy; Authorization Number 349/2016-PR date of approval: 1 April 2016)).

Four-week-old male C57BL/6J (B6) mice, purchased from Harlan Laboratories (San Pietro al Natisone Udine, Italy) were housed in a room with controlled temperature and dark–light cycles, with free access to water and food. After acclimatization (1 week), the animals were weighed and divided into three groups. (1) Lean group—control animals were fed the standard diet (STD; 4RF25 Mucedola, Milan, Italy) for 16 weeks; (2) High-fat diet (HFD) group—obese animals fed HFD (PF4215, Mucedola, Milan, Italy) for 16 weeks. (3) HFD-P group—obese animals fed HFD supplemented with pistachio for 16 weeks. HFD-P was custom designed and prepared by Mucedola S.r.l (PF4215/C; R&S 34/16). It was obtained by substituting 20% of the caloric intake from HFD with pistachio (180 g/kg of HFD). The HFD and HFD-P were stored in vacuum containers at 4 °C. The energy densities of the diets are shown in Table 2.

Pistachio nuts belong to *Pistacia vera* L. species and were purchased by Pistachio Valle del Platani Association and Pistacchio di Raffadali (Agrigento-AG, Sicily, Italy). As previously described [31], during the 16 weeks of the experiment, changes in body weight and food-intake were measured weekly and results from the different groups of animals were compared. At the end, the animals were sacrificed by cervical dislocation; the blood was collected immediately by intracardiac puncture, and the plasma was isolated by centrifugation at 3000 rpm at 4 °C for 15 min and stored at −80 °C, until analysis. The liver, adipose tissue, and small intestine were rapidly removed; a part of each tissue was fixed in 4% neutral formalin solution for histological analysis and another part was stored at −80 °C for biomolecular analysis. Five-centimeter segments of the small intestine were taken for the Ussing chamber assays.

### 4.2. Plasma Biomarker Analysis

IL-1β and TNF-α were quantified by a commercial ELISA Kit (Cloud-Clone Corp, Wuhan, Hubei), based on the manufacturer’s instructions. The levels of triglyceride and total cholesterol in the serum were evaluated by using the automatic biochemical analyzer (ILab 600, Instrumentation Laboratory, Milano, Italy).

### 4.3. Liver and Adipose Tissues Histology and Immunohistochemistry

Hepatic, visceral (epididymal), and subcutaneous white adipose tissues (WAT) were fixed with 4% formaldehyde solution for 24 h and embedded in paraffin. Then, 5 μm sections were prepared and stained with hematoxylin and eosin (H&E) for morphological examination. The number of liver inflammatory foci was calculated by counting the inflammatory cell aggregates in the hepatic lobules, per 5 random fields at a magnification of 20×. Hepatic inflammatory foci were defined as aggregates of inflammatory cells that accumulate in the liver during chronic inflammation [86,87]. The number of adipocytes per microscopical field (density) was determined at a magnification of 20×. The mean surface area of the adipocytes (µm^2^) was calculated using the image analyzer software (Visilog 6, Courtaboeuf, France). Each adipocyte was manually delineated and 700–1000 adipocytes per condition were assessed.

Images of the H&E liver and WAT sections were captured using an optical microscope (Leica DMLB, Meyer instruments, Houston, Texas) equipped with a DS-Fi1 camera (Nikon, Florence, Italy), and were analyzed at 10× and 20× magnification.

For the immunohistochemistry, deparaffinized sections were treated with 3% hydrogen peroxide to inactivate the endogenous peroxidase followed by a rinse in PBS for 5 min. Subsequently, the sections were incubated with the primary antibody Mac-2 at 4 °C overnight (1:2800, Cedarlane, Ontario, Canada CL8942AP). After PBS washing, the sections were incubated with the biotinylated secondary antibody (Anti-Mouse IgG/Rabbit IgG) (1:400, Vector Laboratories, BA-4001) for 30 min. Histochemical reactions were performed using the Vector’s Vectastain ABC Kit (Vector Laboratories, Burlingame, CA, USA) and diaminobenzidine as a substrate (Sigma, Milano, Italia). Crown-like structures (CLS) were counted as a measure of adipose tissue inflammation and were expressed as number of CLS/10,000 adipocytes.

### 4.4. Reverse Transcription Polymerase Chain Reaction (RT-PCR)

RNA was extracted from liver, epididymal, and subcutaneous adipose tissue, using the RNeasy plus Mini Kit (Qiagen, Valencia, CA, USA), according to the manufacturer’s protocol. The extraction from adipose tissues was performed after a preliminary step of lysis using Triazol. Two nanograms of the total RNA were used for cDNA synthesis with High Capacity cDNA Reverse Transcription (Applied Biosystems, Waltham, MA, USA). The target cDNA was amplified using genetic-specific primers, as listed in Table 3. The amplification cycles included denaturation at 95 °C for 45 s, annealing at 52 °C for 45 s, and elongation at 72 °C for 45 s. After 40 cycles, the PCR products were separated by electrophoresis on a 1.8% agarose gel for 45 min at 85 V. The gels were stained with 1 mg/mL ethidium bromide and visualized with ultraviolet (UV) light, using E-Gel GelCapture (Thermo Fisher Scientific, Monza, Italy). The expression levels of the gene targets, normalized to the endogenous reference (β-actin), were analyzed using the E-Gel GelQuant Express Analysis Software (Thermo Fisher Scientific, Monza, Italy).

### 4.5. Gut Microbiota Composition

Six hours before the sacrifice, the mice were kept individually in a clean cage without food and the stool samples were collected from each mouse for gut microbiota analysis, using an autoclaved tube. Bacteria DNA was extracted from stool samples (200 mg per mouse) using the QIAamp DNA Stool Handbook kit (QIAGEN, Milan-Italy), following the manufacturer’s protocol. The extracted DNA was used for the metagenomic study carried out by the BMR Genomics company s.r.l. (Padova, Italy).

For the NGS sequencing, the V3–V5 regions of the 16S rRNA gene were amplified. After confirming that all V3–V5 amplicons had good levels of concentration, purity, and integrity, a massive sequencing was carried out utilizing the Illumina MiSeq platform (San Diego, CA, USA). Reference-based UCLUST algorithm (Qiime1.9.1) was used to pick the OTUs at 97% of similarity against Greengenes v13.8 database. OTUs were collected in the biom file and filtered at 0.005% abundance to eliminate spurious OTUs that were present at a low frequency.

### 4.6. Ussing Chamber Measurements

Intestinal barrier integrity was evaluated in an Ussing chamber system. A segment of small intestine was excised from freshly sacrificed mice and transferred to an ice-cold oxygenated Krebs solution containing (mM) NaCl 119, KCl 4.5, MgSO4 2.5, NaHCO3 25, KH2PO4 1.2, CaCl2 2.5, and glucose 11.1. The segment was cut longitudinally along the mesenteric border and mounted in an Ussing chamber. The Ussing chambers contained a hydrated mixture of 5% CO^2^/95% O^2^ (v/v). The Ussing chamber system were filled with 10  mL Krebs solution, maintained at 37  °C, and continuously bubbled with the short-current (Isc), i.e., the current generated by the ionic transport through the epithelium. The transepithelial potential difference was continuously monitored under open circuit conditions, using a DVC 1000 amplifier (DVC 1000, World Precision Instruments, Sarasota, FL, USA) and was recorded through filled agar electrodes. The conductance was calculated according to Ohm’s law, using the potential difference and current (Isc) values. Tissues whose conductance increased during the course of the experiment (calculated every 15 min) were considered damaged and were excluded from the data analysis. 

### 4.7. Statistical Analyses

Results are shown as means ± the standard error of the mean (SEM). The letter ‘n’ indicates the number of animals. Statistical analyses were performed using the Prism Version 6.0 Software (Graph Pad Software, Inc., San Diego, CA, USA). The comparison between the groups was performed by Analysis of Variance (ANOVA), followed by Bonferroni’s post-test. A *p*-value ≤ 0.05 was considered statistically significant. 

## 5. Conclusions

The present study demonstrated that chronic intake of pistachio exerts beneficial effects in obese mice by alleviating inflammation in adipose tissues and liver, and impacting the gut microbiome composition. In particular, it enhances the abundance of beneficial bacteria genera, such as *Lactobacillus*, *Dorea*, *Allobaculum*, and inhibited the growth of bacteria associated with obesity-related comorbidities and inflammation, such as *Desulfovibrio* and *Bilophila*.

## Figures and Tables

**Figure 1 ijms-21-00365-f001:**
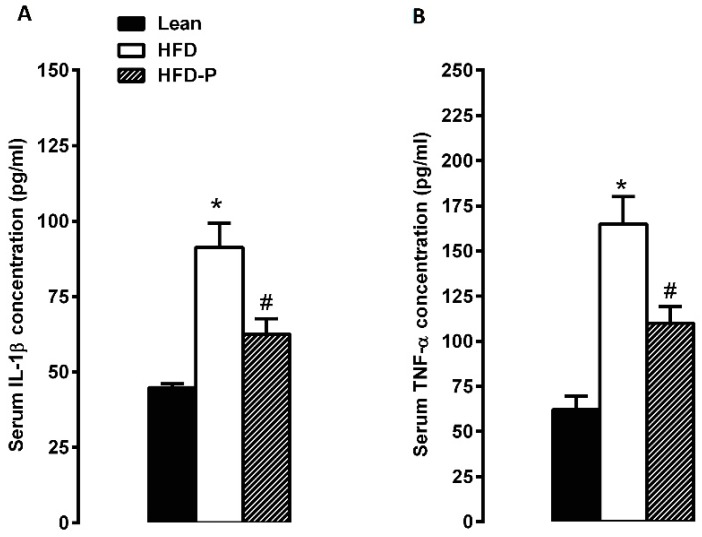
Effects of pistachio consumption on pro-inflammatory cytokines. Serum circulating levels of IL-1β (**A**) and TNF-α (**B**) in the lean, HFD, and HFD-P groups. Data are expressed as mean ± SEM; (*n* = 8/group). * *p* < 0.05 compared with lean; # *p* < 0.05 compared with HFD.

**Figure 2 ijms-21-00365-f002:**
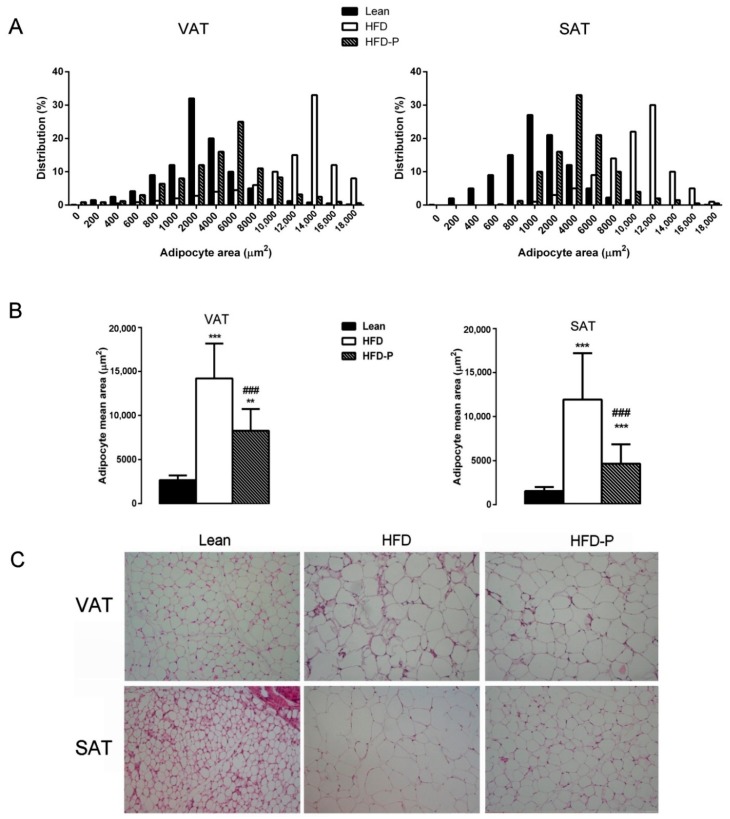
Effects of pistachio consumption on adipocyte morphology. (**A**) Adipocyte size distribution (%) and (**B**) adipocyte mean area (μm^2^) of the epididymal visceral adipose tissues (VAT) and subcutaneous adipose tissue (SAT) in lean, HFD, and HFD-P mice. (**C**) Adipose tissue staining (H&E staining, magnification 10×) in the lean, HFD, and HFD-P mice. Data are expressed as mean ± SEM; (*n* = 8/group). Compared to the lean mice (** *p* < 0.01; *** *p* < 0.001); Compared to the HFD mice (### *p* < 0.001).

**Figure 3 ijms-21-00365-f003:**
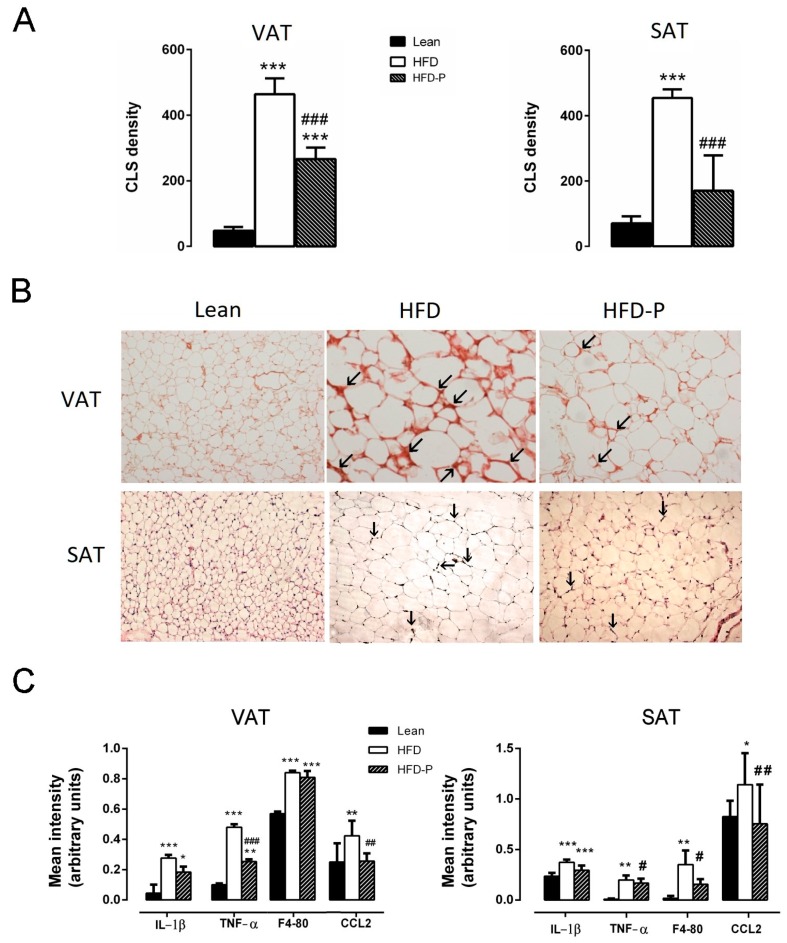
Effects of pistachio consumption on Crown Like Structures CLS density. (**A**) Representative results of the density of MAC-2 positive CLS stained in epididymal visceral adipose tissue (VAT) and subcutaneous adipose tissue (SAT) of the three groups of animals (CLS number/10.000 adipocytes). (**B**) VAT and SAT immunohistochemistry (IHC) analysis for MAC-2 positive macrophages forming CLS (arrows) in the lean, HFD, and HFD-P animals (magnification 10×). (**C**) Effect of Pistachio consumption on IL-1β, TNF-α, F4-80, and CCL2 mRNA expression in VAT and SAT of the lean, HFD, and HFD-P mice. Data are expressed as mean ± SEM; (*n* = 8/group). * *p* < 0.05 compared to the lean mice (* *p* < 0.05; ** *p* < 0.01; *** *p* < 0.001); # *p* < 0.05 compared to the HFD mice (# *p* < 0.05; ## *p* < 0.01; ### *p* < 0.001).

**Figure 4 ijms-21-00365-f004:**
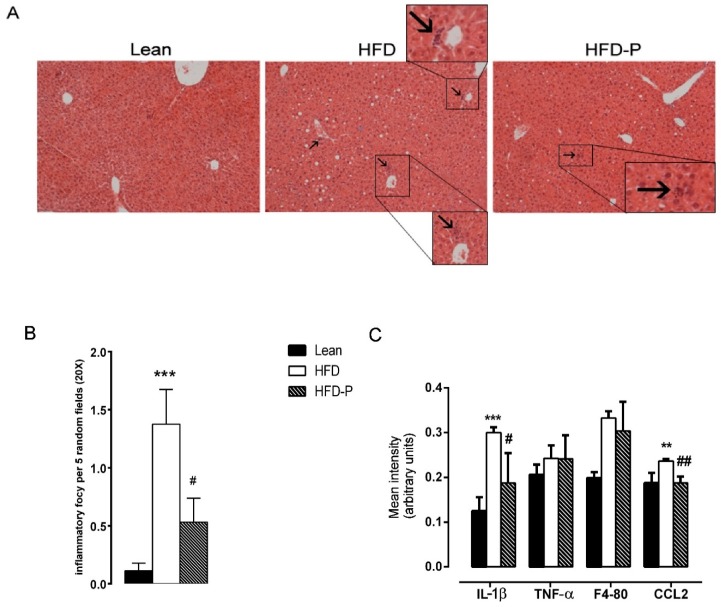
Effect of pistachio consumption on liver inflammation. (**A**) Liver histology of the lean, HFD, and HFD-P mice was examined by H&E staining. Arrows indicate the points of inflammatory foci (magnification 10×). (**B**) Quantification of inflammatory foci per 5 random fields under 20× magnification. (**C**) mRNA levels of IL-1β, TNF-α, F4-80, and CCL2 in the livers of the lean, HFD, and HFD-P mice (**B**). Data are represented by the means ± SEM. (*n* = 8/group). * *p* < 0.05 compared to the lean mice (** *p* < 0.01; *** *p* < 0.001); # *p* < 0.05 compared to the HFD mice (# *p* < 0.05; ## *p* < 0.01).

**Figure 5 ijms-21-00365-f005:**
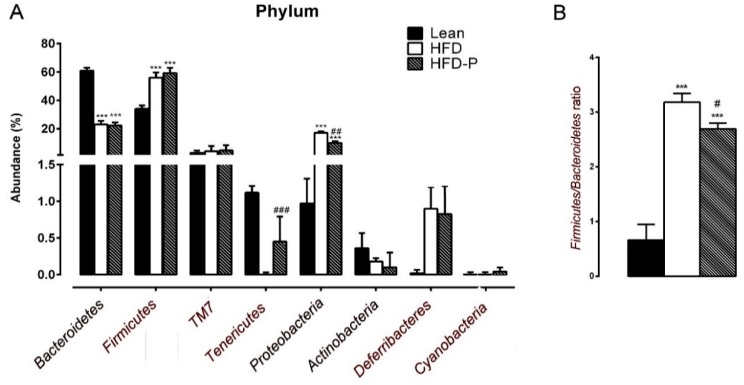
16S rDNA sequencing of bacterial DNA in the feces of the lean, HFD, and HFD-P mice, in order to discriminate the intestinal microbial profile. (**A**) Graphic representation of the relative abundance (%) of the gut microbiota phyla composition of the three groups of animals. (**B**) Ratio of *Firmicutes* to *Bacteroidetes* in the lean, HFD, and HFD-P. Data are expressed as mean ± SEM; (*n* = 8/group). (*** *p* < 0.001); hash denotes significant difference when compared to the HFD group (# *p* < 0.05; ## *p* < 0.01; ### *p* < 0.001).

**Figure 6 ijms-21-00365-f006:**
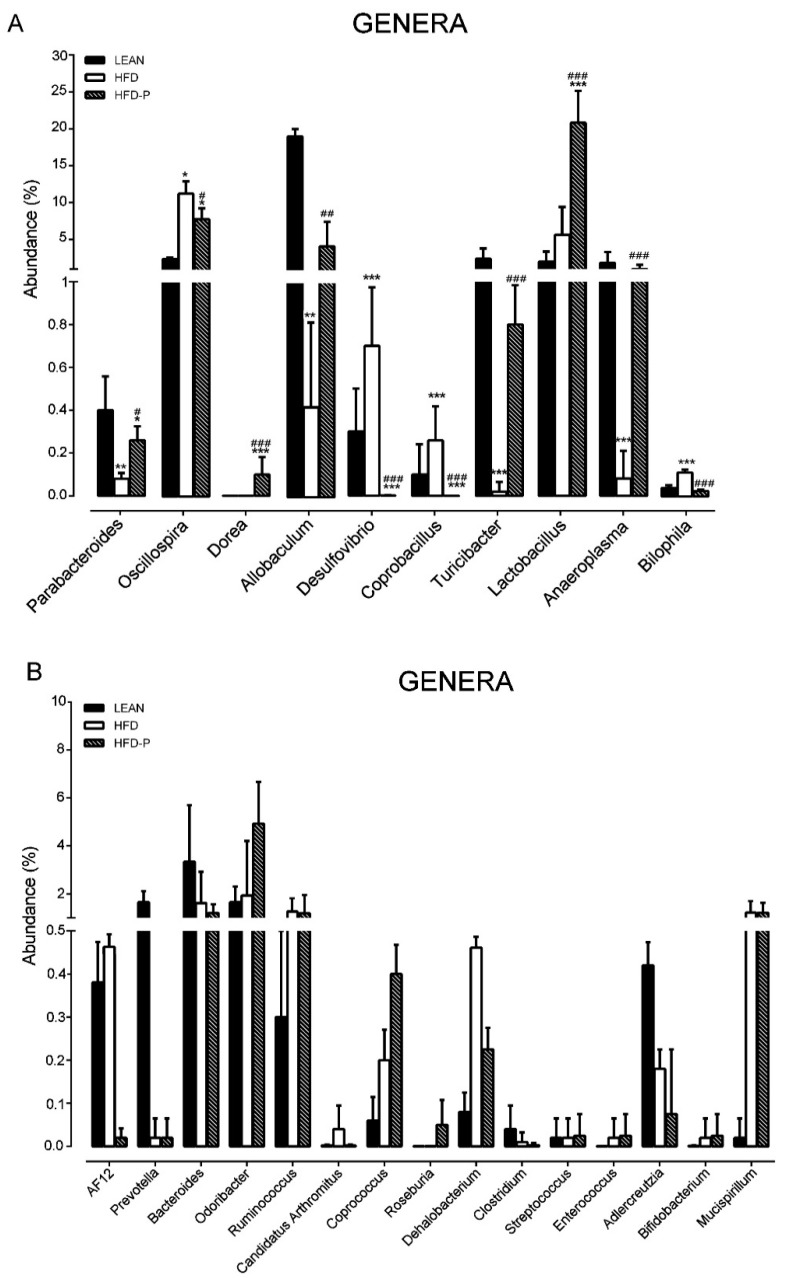
Genus level taxonomic distributions of the microbial communities in the feces of the lean, HFD, and HFD-P mice. (**A**) Genera abundance (%) was significantly modified by the pistachio intake. (**B**) Genera abundance (%) was not modified by the pistachio intake. Data are expressed as means ± SEM; (*n* = 8/group). (* *p* < 0.05; ** *p* < 0.01; *** *p* < 0.001); hash denotes significant difference compared to the HFD group (# *p* < 0.05; ## *p* < 0.01; ### *p* < 0.001).

**Figure 7 ijms-21-00365-f007:**
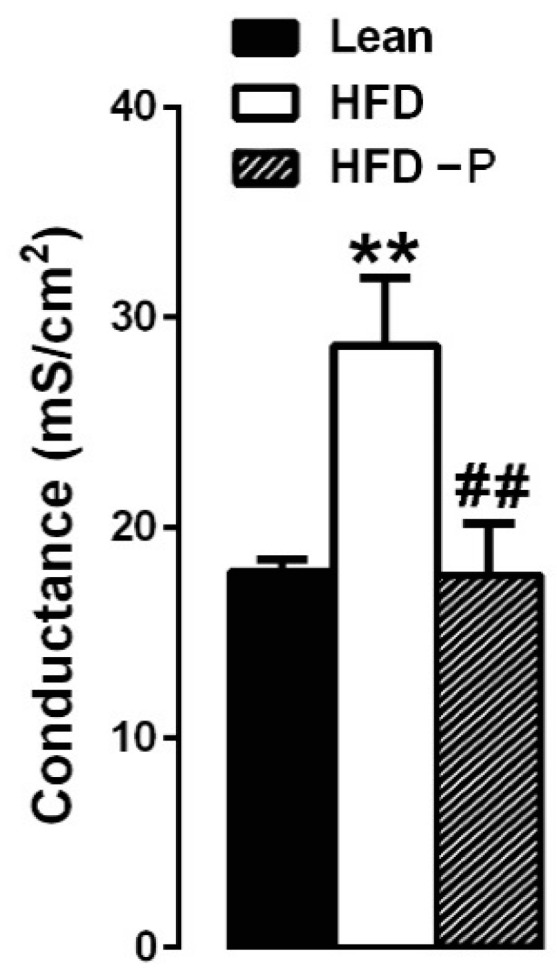
Effect of pistachio consumption on the conductance of isolated duodenal sections from the lean, HFD, and HFD-P mice through the Ussing chambers technique. Data are expressed as mean ± SEM; (*n* = 8/group). (** *p* < 0.01); hash denotes significant difference compared with the HFD (## *p* < 0.01).

**Table 1 ijms-21-00365-t001:** Effects of pistachio consumption on high-fat diet (HFD)-related dysmetabolisms.

	Lean	HFD	HFD-P
Body weight (g)	32.3 ± 0.9 g	46.2 ± 1.1 g *	46 ± 1.2 g *
Food Intake (g)	4.05 ± 0.2 g	3.4 ± 0.08 g	3.3 ± 0.07 g
Triglycerides (mg/dL)	82 ± 4.5 mg/dL	119 ± 5.5 mg/dL *	93.1 ± 5.1 mg/dL ^#^
Cholesterol (mg/dL)	100 ± 5 mg/dL	192 ± 4 mg/dL *	150 ± 4 mg/dL ^#^

Body weight, food intake, triglyceride and cholesterol plasma concentrations of lean, HFD, and HFD supplemented with pistachios (HFD-P) animals at the end of the experimental period. Data are expressed as mean ± SEM (*n* = 8/group). * *p* < 0.05 compared with lean; ^#^
*p* < 0.05 compared with HFD.

**Table 2 ijms-21-00365-t002:** Composition and energy densities of the STD, HFD, and HFD-P groups.

Ingredient (g/kg)	STD	HFD	HFD-P
Total Energy, Kcal/g	3.5	6	6
Protein, %	20	20	20
Carbohydrate, %	70	20	20
Fat, %	10	60	60

STD—standard diet. HFD—high fat diet. HFD-P—HFD supplemented with pistachio.

**Table 3 ijms-21-00365-t003:** Oligonucleotide sequence of primers for RT-PCR.

Gene	Forward Primer	Reverse Primer	Size (bp)
IL-1β	5′-CAGGATGAGGACATGAGCACC-3′	5′-CTCTGCAGACTCAAACTCCAC-3′	450
TNF-α	5′-AGCCCACGTCGTAGCAAACCA-3′	5′-GCAGGGGCTCTTGACGGCAG-3′	260
F4-80	5′-GCCACGGGGCTATGGGATGC-3′	5′-TCCCGTACCTGACGGTTGAGCA-3′	360
CCL2	5′-TCTGTGCTGACCCCAAGAAGG-3′	5′-TGGTTGTGGAAAAGGTAGTGGAT-3′	183
*β-actin*	5′-GGATCCCCGCCCTAGGCACCAGGGT-3′	5′-GGAATTCGGCTGGGGTGTTGAAGGTCTCAAA-3′	289

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
