# Peer review of "Pistachio Consumption Alleviates Inflammation and Improves Gut Microbiota Composition in Mice Fed a High-Fat Diet"

_ijms, 2020, doi:10.3390/ijms21010365_

Round 1
Reviewer 1 Report
The manuscript is well written, original and fulfills the aims of the Journal.
It just needs a fair revision of English style.
e.g. use of comma at line 88 “As previously reported [27,31], after 16 weeks on HFD, please remove comma mice showed a significant increase in…. “.
Line 120: “As shown in Figure 3, more crown-like structures were detected in HFD mice, compared 120 to lean animals please rephrase " as compared with… ".
Author Response
Response to Reviewer 1 Comments
We thank you for your suggestions and comment.
Point 1. Line 88 “As previously reported [27,31], after 16 weeks on HFD, please remove comma mice
showed a significant increase in…. “.
Response 1. As you suggested, comma was removed.
Point 2. Line 120: “As shown in Figure 3, more crown-like structures were detected in HFD mice, compared 120 to lean animals please rephrase " as compared with… ".
Response 2. The sentence was rephrased as you suggested.
Reviewer 2 Report
Tezro et al. found that regular pistachio consumption improved inflammation in obese mice. The positive effects could be related to positive modulation of the microbiota composition. Overall the study is well designed and the outcomes appropriate. The experimental section is descriptive with the detailed discussion of the results. The manuscript has a good amount of citations to the references.
Only a few minor concerns:
Some text was shown in red-color, such as in line 54 and 485. Abbreviation list was not complete. What about HFD, TNF, etc… In addition, what is CLS stands for (line 119 and 360)? Alternatively, you may delete this list. Page 7 line 166, text was wrong in bold font.” Methods: which part of small intestine was dissected? (line 319) Reference style was inconsistent, with irregular upper and lower case shown.
Author Response
Response to Reviewer 2 Comments.
We thank you for your suggestions and comment.
Point 1. Text was shown in red-color, such as in line 54 and 485.
Response 1. Thanks for noticed. The text in red color was converted in black.
Sede di Viale delle Scienze Ed.16 - 90128 Palermo mail:[email protected]
Centr.no 091.23897111 – Amm.ne 091.23897201-202-207-208-210 Fax 091.6577210 web: http://portale.unipa.it/stebicef
DIPARTIMENTO DI SCIENZE E TECNOLOGIE BIOLOGICHE CHIMICHE E FARMACEUTICHE (STEBICEF)
Point 2. Abbreviation list was not complete. What about HFD, TNF, etc… In addition, what is CLS stands for (line 119 and 360)? Alternatively, you may delete this list.
Response 2. Thanks for the suggestion. The list has been done.
Point 3. Page 7 line 166, text was wrong in bold font.
Response 2. Thanks for noticed. The bold font has been eliminated.
Point 4. Methods: which part of small intestine was dissected? (line 319).
Response 4. The small intestine dissected was duodenum.
Point 5. Reference style was inconsistent, with irregular upper and lower case
shown.
Response 5. Thanks for noticed. The reference style has been corrected.